# Association between Postpartum Nutritional Status and Postpartum Depression Symptoms

**DOI:** 10.3390/nu11061204

**Published:** 2019-05-28

**Authors:** Yu-Hung Lin, Chiao-Ming Chen, Hui-Min Su, Shu-Ci Mu, Mei-Ling Chang, Pei-Yin Chu, Sing-Chung Li

**Affiliations:** 1Department of Obstetrics and Gynecology, Shin-Kong Wu Ho-Su Memorial Hospital, Taipei 11101, Taiwan; yuhung453@gmail.com; 2School of Medicine, Fu-Jen Catholic University, New Taipei City 24205, Taiwan; musc1006@yahoo.com.tw; 3Department of Obstetrics and Gynecology, Taipei Medical University, Taipei 11031, Taiwan; 4Department of Obstetrics and Gynecology, National Taiwan University Hospital, Taipei 10041, Taiwan; 5Department of Food Science, Nutrition, and Nutraceutical Biotechnology, Shih Chien University, Taipei 10462, Taiwan; charming@g2.usc.edu.tw (C.-M.C.); mlchang@g2.usc.edu.tw (M.-L.C.); michelle021883@gmail.com (P.-Y.C.); 6Physiology, College of Medicine, National Taiwan University, Taipei 10051, Taiwan; hmsu1203@ntu.edu.tw; 7Department of Pediatrics, Shin-Kong Wu Ho-Su Memorial Hospital, Taipei 11101, Taiwan; 8School of Nutrition and Health Sciences, College of Nutrition, Taipei Medical University, 250 Wu-Hsing Street, Taipei 11031, Taiwan

**Keywords:** confinement, postpartum nutritional status, postpartum depression symptoms, riboflavin, n-3 fatty acid

## Abstract

Taiwanese women may practice traditional confinement after childbirth, and no study has investigated the nutritional status and the effects of postpartum depression on such women. The aim of this study was to investigate the association between nutritional status and postpartum depression at 6–8 weeks postpartum. A cross-sectional study was conducted on postpartum women who returned to the obstetrics and gynecology clinic for routine examination from January 2016 to September 2017. A total of 344 women received assessments based on the Edinburgh Postnatal Depression Scale (EPDS). An EPDS score of ≥10 indicated the presence of postpartum depressive symptoms (PPDS). A total of 97 women without such symptoms and 23 with PPDS completed nutritional parameter analyses and questionnaires. The results showed that the prevalence of postpartum depression (PPD) was 8.4%. The proportion was 70% for those who practiced confinement at home, significantly higher than for those in the non-PPDS group (45%). The overall psychological stress score was significantly higher and the postpartum care satisfaction score was significantly lower in those with PPDS compared to those without. In terms of nutritional biomarkers, the plasma riboflavin levels in the PPDS group were significantly lower than those in their symptomless counterparts (13.9%). The vitamin D insufficiency and deficiency rates in the non-PPD and PPDS groups were 35%, 41%, 48%, 26%, respectively. However, compared with those in the non-PPDS group, those with PPDS had significantly higher ratios of Σn-6/Σn-3, C20:3n-6/C18:3n-6, and C20:4n-6/(C20:5n-3 + C22:6n-3) (by 8.2%, 79.7%, and 8.8%, respectively), whereas they had lower ratios of C22:6n-3/C22:5n-6 (by 15.5%). Higher plasma riboflavin and erythrocyte C16:1n-9, C24:1n-9, C18:3n-6, and C20:5n-3 levels and lower Σn-6 fatty acid and C22:5n-6 levels decreased the risk of PPD after type of confinement, overall mental stress scores, and postpartum care satisfaction scores were adjusted for the logistic regression analysis. In conclusion, the plasma riboflavin level and erythrocyte fatty acid composition are potentially major contributors to PPD development.

## 1. Introduction

Postpartum depression (PPD) is a debilitating mental disorder with a prevalence of between 3.5% and 63.3% in Asian countries [1]. An international study exploring levels of PPD showed European and Australian women had the lowest levels of PPDS, women in the USA fell at the midpoint, and women in Asia and South America had the highest depressive symptom scores [2]. The national prevalence of minor/major and major PPDS was found to be 8.46% and 8.69%, respectively among Canadian women [3]. PPD in women can manifest as sleep disorder, mood swings, sadness and crying, fear of injury, loss of appetite, serious concerns about their child, a lack of interest in daily activities, and even suicide and death [4]. The risk factors of PPD are complicated; they include psychiatric (e.g., a previous history of depression and anxiety), obstetric (e.g., postpartum complications), biological (e.g., young age, glucose metabolism disorders), hormonal (e.g., oxytocin and estrogen imbalances), social (e.g., a lack of familial support or financial support), and lifestyle (e.g., food intake patterns, sleep status, exercise, and physical activity) factors [5,6,7,8,9,10].

The Edinburgh Postnatal Depression Scale (EPDS) is the most commonly used screening scale for PPD symptoms (PPDS) because it is reliable, has been sufficiently validated, and is often more practical and cost-effective than other methods for the comprehensive screening of PPD risks [11]. The Chinese version of the EPDS has appropriate reliability and validity for screening for PPD among Taiwanese women [12]. The prevalence of PPD screened using the EPDS in Taiwan was 21%, 42%, and 39% in 2004, 2007, and 2013, respectively [13,14,15]. The results of these studies also indicated that cesarean section, postpartum complications, a previous history of depression, unplanned pregnancy, low income, and a lack of familial support increase PPDS risk. 

Recent evidence from studies has shown that malnutrition also contributes to PPD development, including deficiencies in *n*-3 polyunsaturated fatty acids (PUFA), B vitamins, vitamin D, and trace minerals [16]. Taiwan has a traditional custom involving the confinement of the mother after childbirth. During this period, mothers are supposed to eat nutritious foods and herbs to aid their recovery. The relationship between the nutritional status of postpartum women in confinement and postpartum depression has never been investigated. Thus, this study investigated the association between nutritional status and postpartum depressive symptoms at 6–8 weeks postpartum. 

## 2. Participants and Methods

### 2.1. Study Participants

A cross-sectional study was conducted on postpartum women who returned to the obstetrics and gynecology clinic of Shin Kong Wu Ho-Su Memorial Hospital for a routine postpartum examination and who met the inclusion criteria. They were recruited from January 2016 to September 2017. The inclusion criteria for the participants were postpartum women (6–8 weeks) who were healthy and without diseases, did not smoke or drink alcohol, and who had infants born at term who were not diagnosed with any diseases. Any participant with a congenital disease (e.g., thalassemia, thyroid abnormality, and diabetes), multiple births, a premature birth, an acute infection (e.g., a cold, mastitis) or who had ever been diagnosed with a mental disorder (including schizophrenia, depression, and bipolar disorder) were excluded.

The Ethics Committee of Shin Kong Wu Ho-Su Memorial Hospital approved this study according to the International Organizations of Medical "International Ethical Guidelines for Biomedical Research Involving Human Subjects” and the ethical principles of the Declaration of Helsinki (20150810R). We obtained written informed consent from the participants before the study procedures were performed. 

### 2.2. PPDS Screening and Questionnaires

PPDS screening in this study was conducted using the Chinese version of the EPDS, which has good reliability and validity and is widely used in Taiwan [12]. A total of 344 women received EPDS assessments; an EPDS score of ≥10 was defined as the presence of PPDS and an EPDS score of <10 indicated no PPSS. In addition, participants were asked to answer a questionnaire designed to obtain the following sociodemographic information: age, education level, occupation, parity, baby’s sex, type of delivery, baby’s main caregiver, breastfeeding status, perinatal and lactation nutritional supplements, type of postpartum confinement, self-perceived psychological stress, and postpartum care satisfaction scores. The self-perceived psychological stress item consisted of seven yes-or-no questions (no = 0, yes = 1), with all scores added to obtain the overall stress score. Postpartum care satisfaction was assessed by using a visual analog scale (VAS). The VAS was altered to a numeric scale from 0 to 10. Pre-pregnancy body weight (kg) and gestational weight gain (kg) were determined from self-reported data; current body weight (kg), and height (m) were measured with an electronic health scale (WB-380H, TANITA, Japan).

### 2.3. Blood Sample Collections

Blood samples (6 mL) were collected from a vein of one arm into vacutainers with and without anticoagulant (EDTA) by a trained technician. The serum was collected after centrifugation at 1400 g for 10 min at 4 °C. Half of the aliquot was stored at −80 °C for vitamin analysis and the other half kept at 4 °C and immediately sent to Zhong-yi Medical Laboratory for analysis. After the plasma and buffy coat layer were removed, the remaining red blood cells (RBCs) were washed three times with normal saline (0.9% NaCl) that was approximately twice the volume of the RBCs and stored at −80 °C for RBC fatty acid analysis.

### 2.4. Biochemical Analyses

The complete blood count was determined using a hematology analyzer (XN10, Xysmex, Japan), and serum 25-hydroxyvitamin D [25(OH)D] and ferritin levels were determined using an electrochemiluminescence immunoassay (Roche Cobas 6000, Switzerland); the interday coefficients of variation (CVs) were 8.64%, 3.97%, and 2.25%, respectively. Plasma α-tocopherol and retinol levels were determined using a reverse-phase high performance liquid chromatography (HPLC) method according to Bieri et al. [17]. Briefly, α-tocopherol and retinol in plasma were extracted with hexane and then quantified using an HPLC system (Hitachi, Japan) equipped with a LiChroCART C-18 column (4 × 250 nm^2^, 4 μm; Perkin-Elmer, West Lafayette, IN, USA) and a UV/VIS detector (Hitachi, Japan). The concentration of plasma α-tocopherol and retinol was calculated with the assistance of a standard curve constructed using pure α-tocopherol and retinol (Sigma-Aldrich); the intraday CVs were 3.3% and 3.1%, and the interday CVs were 10.6% and 11.7%, respectively. Plasma riboflavin was assessed using an LSBio vitamin B2/riboflavin enzyme-linked immunosorbent assay (ELISA) kit (Lifespan Biosciences, Seattle, WA, USA); the intraday CV was 6% and the interday CV was 13.7%.

### 2.5. Erythrocyte Fatty Acid Profiles Analysis

Aliquots of 200 μL of RBC suspension were pipetted into a Teflon-sealable Sovirel tube containing 3 mL of methanol/chloroform (2:1 *v*/*v*) and 100 μL of 13:0 internal standard that was filled with nitrogen gas to avoid fatty acid (FA) oxidation. After these were mixed and left to react for 4 hours at 4 °C, chloroform and 0.7% NaCl were added. The lower phase was aspirated and dried under nitrogen flow and then mixed with methanol/dichloromethane (3:1 *v*/*v*) and acetyl chloride to prepare FA methyl esters (FAMEs). FAMEs were extracted by hexane and analyzed on an Agilent Technologies 7820A gas chromatograph using flame ionization detection. The FA peaks were identified from external standard runs through the comparison of the retention times with those of an authentic standard mixture of 68A (Nu-Chek Prep, Elysian, MN, USA), 37 FAME, PUFA2, and PUFA3 (all from SUPELCO, Bellefonte, PA, USA). The FA composition of each lipid was expressed with the weight of that lipid as a percentage of the total weight of the carbon 12 to carbon 22 FAs (wt %).

### 2.6. Statistical Analysis

Data are presented as means ± standard deviations (SDs) or as a percentage. Our data were confirmed to have a normal distribution by using a Kolmogorov–Smirnov test. Between-group analyses were conducted using a Student’s t test or a Pearson’s chi-squared test. Correlations were made using a multivariable logistic regression to assess the associations between nutritional biomarkers and PPDS risk. All data analyses were performed using SPSS version 19 (SPSS Inc., Chicago, IL, USA). Differences were considered significant at *p* < 0.05.

## 3. Results

### 3.1. Participant Characteristics

A total of 460 women were eligible for this study. EPDS screening was administered in 344 women after six were excluded for having multiple births and 110 declined to participate. A total of 315 women were without symptoms (EPDS score < 10), and the remaining 29 were deemed to have PPSD (EPDS score ≥ 10), resulting in a prevalence of 8.4%. Blood samples and completed questionnaires of 120 participants (97 with and 23 without PPDS) were obtained (Figure 1).

The characteristics of the participants are listed in Table 1, which provides proportions and means. The general demographic information for the two groups (with and without PPDS) was comparable. No significant differences were noted in pre-pregnancy body mass index (BMI), gestational weight gain, current BMI, years of education, occupation, parity, baby’s sex, type of delivery, baby’s caregiver, breastfeeding, and nutritional supplements during the perinatal and lactation periods. Overall, 70% of the participants in the PPDS group practiced confinement at home, a significantly larger proportion than those who did in the non-PPDS group (45%). The PPDS group also had more self-perceived psychological stress, especially in terms of ability to care for the baby and support from the baby’s father and their families. The overall psychological stress and postpartum care satisfaction scores of the PPDS group were significantly higher and lower, respectively, compared with those of the women in the non-PPDS group.

### 3.2. Biochemical Analyses

The nutritional biomarkers of the postpartum women are shown in Table 2. No significant differences were noted in average of HGB (hemoglobin), ferritin, 25(OH)D3, retinol, and α-tocopherol between the two groups, except for riboflavin. The plasma riboflavin level in the PPDS group was significantly lower than in the non-PPDS group (by 13.9%). Vitamin D insufficiency and deficiency were defined as serum 25 (OH)D3 levels of <30 ng/mL and <20 ng/mL, respectively [18]. The average concentrations of 25 (OH)D3 in the two groups were insufficient. We noted that postpartum women generally had insufficient vitamin D status, and the vitamin D insufficiency and deficiency rates in the non-PPDS and PPDS women were 35%, 41%, 48%, and 26%, respectively.

Erythrocyte FA compositions are shown in Table 3. Our data indicate that total saturated FAs, monounsaturated FAs (MUFAs), and n-6 PUFAs were not significantly different between the two groups, except for n-3 PUFAs. The PPDS group had significantly lower total n-3 PUFA levels, especially C20:5n-3 (eicosapentaenoic acid, EPA), and C22:6n-3 (docosahexaenoic acid, DHA), which decreased by 16.8% and 4.5%, respectively. Moreover, women with PPDS had significantly lower C16:1n-9, C24:1n-9, and C18:3n-6 levels and higher C20:3n-6 (dihomo-γ-linolenic acid, DGLA) and C22:5n-6 (n-6 docosapentaenoic acid, n-6DPA) levels. Furthermore, compared with the values of the non-PPDs group, the PPDS group had significantly elevated ratios of Σn-6/Σn-3, C20:3n-6/C18:3n-6, and C20:4n-6/(C20:5n-3 + C22:6n-3) by 8.2%, 79.7%, and 8.8%, respectively, whereas they had a decreased ratio of C22:6n-3/C22:5n-6 by 15.5%.

### 3.3. Association of Nutritional Biomarkers with Postpartum Depression

Correlations between nutritional biomarkers and PPDS are shown in Table 4. We fit two multivariate models to the data. In Model 1, no covariates were adjusted, and in Model 2, the type of postpartum confinement, overall psychological stress scores, and postpartum care satisfaction scores were adjusted. The data revealed that the plasma riboflavin level was associated with a decreased risk of having PPDS (odds ratio [OR]: 0.747; 95% CI: 0.566–0.987; *p* = 0.040). A similar trend was noted after covariates were adjusted (OR: 0.684; 95% CI: 0.504–0.930; *p* = 0.015). By contrast, no significant associations were observed between PPDS and serum 25(OH)D3, retinol, and α-tocopherol levels (all *p* > 0.05). 

The associations between PPDS and erythrocytes FA composition are shown in Table 5. The data indicate that the total MUFA levels significantly decreased PPDS risk, whereas n-6 PUFA levels increased the risk after the covariates were adjusted. In terms of the MUFA profiles, C16:1n-9 and C24:1n-9 levels had a negative correlation with PPDS (OR = 0.504, 95% CI = 0.314–0.808, *p* = 0.004 and OR = 0.967, 95% CI = 0.943–0.990, *p* = 0.006, respectively). In terms of the n-6 PUFA profiles, more long-chain FA C22:5n-6 contributed to an increased PPDS risk (OR = 1.044, 95% CI = 1.001–1.089, *p* = 0.045), whereas C18:3n-6 reduced the risk of PPDS after covariates were adjusted (OR = 0.630, 95% CI = 0.472–0.842, *p* = 0.002). We found that total *n*-3 PUFAs significantly decreased PPDS risk, but no significant difference was observed after the covariates were adjusted. Regardless of whether the covariates were adjusted, the C20:5n-3 level and ratio of C22:6n-3/C22:5n-6 significantly reduced the risk of PPDS, whereas the ratio of C20:3n-6/C18:3n-6 and C20:4n-6/(C20:5n-3 + C22:6n-3) significantly increased said risk.

## 4. Discussion

In total, 120 participants submitted complete data, and 23 of those had PPDS. Although the number of patients in the non-PPDs group was four times higher than in the PPDS group, the data were confirmed to have a normal distribution by using the Kolmogorov–Smirnov test. Therefore, the sample was representative. 

The prevalence of patients with PPD screened by the EPDS in Taiwan was found to be 21%, 42%, and 39% in 2004, 2007, and 2013, respectively [13,14,15]. The incidence of PPDS in this study was thus significantly lower than in these past surveys. We speculate that this reduction could be related to a policy implemented by the National Health Agency. Since 2011, this agency has provided information during pregnancy to women via a maternal health handbook, and the agency has strengthened the implementation of postpartum depression screening and referral services through the provision of health care resource information during pregnancy checkups. 

We found the practice of confinement in postpartum care centers by the non-PPDS group was as high as 55%, compared with 30% in the PPDS group. The non-PPDS group also exhibited lower levels of self-perceived mental stress, higher postpartum care satisfaction scores, and lower PPDS prevalence (12% vs. 27%; *p* = 0.045). We speculate that women might obtain more rest and have reduced anxiety when they practice confinement in a postpartum care center due to the presence of professional care personnel to help them care for their babies and provide postpartum maternal care. 

Few studies have investigated the relationship between vitamin B2 levels and PPD. Miyake et al. investigated the association between the dietary intake of B vitamins and PPD occurrence in a prospective cohort of 865 women. Only an inverse association was identified between vitamin B2 intake and PPD [19]. Based on the Nutrition and Health Surveys in Taiwan, the prevalence rates of normal and marginally deficient statuses for riboflavin were 71.6% and 21.3%, respectively. Among age strata, women aged 19–30 years and 31–50 years had a higher prevalence in terms of having a deficient or marginally deficient status than other age strata [20]. Although no data were available on the riboflavin status of women in the perinatal and postpartum stages, according to the aforementioned surveys, women of childbearing age seem to have poor vitamin B2 nutritional status in Taiwan. Our data are the first to reveal the negative correlation between plasma riboflavin levels and PPDS in Taiwan. Future nutritional educational policies should encourage the consumption of vitamin B2-rich foods in pregnant and postpartum women, which may help to reduce the incidence of PPDS.

Numerous studies have demonstrated that vitamin D deficiency during pregnancy and the postpartum stage increases the risk of PPD [21,22,23,24]. However, this was not evident in our study results. Our results indicate that the ratio of vitamin D insufficiency and deficiency was high in both groups. Thus, we posit that the vitamin D status in these groups was not sufficient to produce a protective effect. Ethnically Chinese people have a tradition of confinement after childbirth. During confinement, women are provided with nutritious foods, including stewed chicken, fish soup, and fried eggs. We speculate that the reasons for the poor vitamin D nutritional status of these participants may be as follows: (1) They had a poor vitamin D nutritional status during prenatal and perinatal periods. We did not investigate the nutritional status of vitamin D before or during pregnancy. Some studies have shown that women of childbearing age have poor vitamin D status in Taiwan. Li et al. investigated the vitamin D status of 3954 individuals without chronic kidney disease aged ≥30 years in northern Taiwan. Their results indicated that the mean 25(OH)D3 concentration was 28.94 ± 10.27 ng/mL, and the prevalence of vitamin D deficiency was highest in participants aged 30 to 39 years (38.4%) [25]. Our data agree with their results. (2) They unintentionally reduced the synthesis of vitamin D in their skin by reducing their exposure to sunlight. Traditionally, women in confinement cannot go outdoors, which may reduce exposure to the sun, thus worsening vitamin D status. Therefore, we suggest that if women are to follow the tradition of confinement, vitamin D supplementation may be required. 

Early observational studies have consistently demonstrated that serum and dietary n-3 PUFA levels are low in patients with depression [26]. Recent studies have also indicated that decreased serum EPA and DHA concentrations in postpartum women were correlated with a higher PPD in China [27,28]. Our results show that the EPA concentration in women with PPDS was significantly lower than in those without PPDS. Logistic regression also revealed that reduced EPA concentrations increase the risk of postpartum depression. The mechanism of n-3 polyunsaturated FAs might decrease the prevalence and improve the symptoms of depression by limiting the production of proinflammatory eicosanoids and cytokines and regulating the production, function and metabolism of serotoninergic neurotransmitters [29,30]. Bove et al. showed that cortical serotonin concentrations were significantly lower in animals pre- and post-natal fed with n-3 PUFA deficient diet, which lead to impairments in neurochemical parameters. Furthermore, n-3 PUFA might influence serotonin neurotransmission acting through the inflammatory pathways [31]. In particular, it has been reported that low n-3 PUFA intake in diets is correlated with the development of depressive and anxiety-like symptoms [32]. The fish oil treatment significantly attenuated the decrease of serotonin levels in brain areas induced by lipopolysaccharide [33]. Therefore, whether the higher n-3 PUFA content contributes to PPD in clinical improvement warrants further research.

Lotrich et al. demonstrated that higher plasma DGLA (C20:3n-6,) levels and the ratio of arachidonic acid (C20:4n-6, AA) divided by EPA + DHA were associated with subsequent depression [34]. Mamalakis et al. investigated the relationship between adipose tissue FA profiles and depression. Their results showed that Beck Depression Inventory scores correlated positively with the adipose tissue C20:3n-6/C18:3n-6 ratio. [35] The higher ratios of AA /(EPA + DHA) and C20:3n-6/C18:3n-6 indicate that the body tends to activate the cyclo-oxygenase (COX) inflammation pathway. The COX pathway converts the body’s primary n-6 PUFA, and arachidonic acid to proinflammatory cytokines such as prostaglandins and prostacyclins. Therefore, we speculate that the PPDS group had a higher Σn-6/Σn-3 ratio, which promoted long-chain n-6 FA synthesis and the production of proinflammatory eicosanoids, thus increasing PPDS. Otto et al. conducted an observational study and demonstrated that functional DHA status, expressed as the ratio of DHA/n-6DPA, was significantly lower in a possibly depressed group compared with a non-depressed group [36]. Our data show that the PPDS group significantly elevated C22:5n-6 levels and reduced the ratio of C22:6n-3 (DHA)/C22:5n-6 (n-6DPA) compared with the non-PPDS group. This result indicates that participants in the PPDS group might also increase depressive symptoms by decreasing functional DHA status. 

In this study, notably, we found that C16:1n-9 (palmitoleic acid) and C24:1n-9 (nervonic acid) levels were significantly lower in the PPDS group. No study has explored the relationship between these two FAs and depression, except for one that found a significant decrease in nervonic acid levels in patients with recent-onset schizophrenia [37]. More research is required in the future to clarify its role in PPD. 

The strength of the present work is that it is the first to investigate the relationship between postpartum nutrition status and PPDS in Taiwan. The results can serve as a reference for nutrition policymakers. However, the study had some limitations. First, this was a cross-sectional study that was unable to clarify the effects of longitudinal nutritional status on PPD development. Second, the sample size was relatively small, and all participants lived in northern Taiwan, which might limit the generalizability of the results. Therefore, further large-sample, multicenter studies are required. Third, we only prioritized examining several nutrients that Taiwanese women are more likely to be deficient in and are related to PPDS development. Studying the relationship of other nutrients and PPDS development is warranted. 

## 5. Conclusions

Plasma vitamin B2 levels and erythrocyte FA composition might have a major effect on PPD development. Our results suggest that the moderate consumption of riboflavin and n-3 FA could have protective effects on PPDS.

## Figures and Tables

**Figure 1 nutrients-11-01204-f001:**
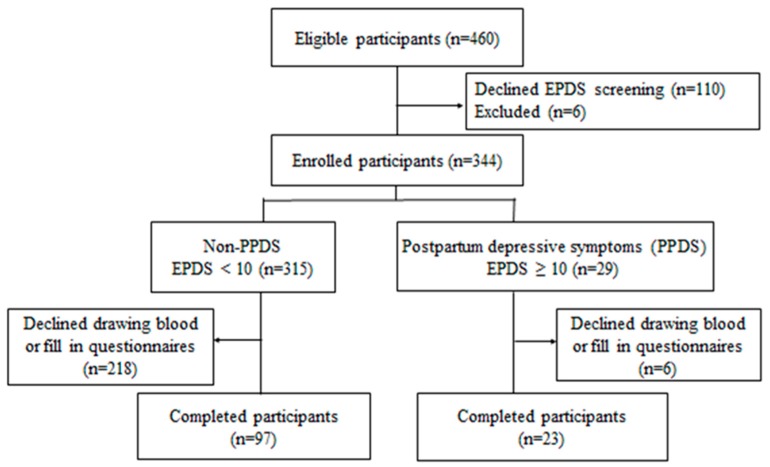
Flowchart of the study.

**Table 1 nutrients-11-01204-t001:** Characteristics of the participants.

	Non-PPDS	PPDS	
	*N* = 97	*N* = 23	
	*n* (%) or mean ± SD		*p* value
**Age (year)**	32.6 ± 4.5	31.6 ± 4.3	0.353
**Anthropometric**			
Pre-pregnancy BMI (kg/m^2^)	21.6 ± 3.2	21.5 ± 3.2	0.853
Gestational weight gain (kg)	12.9 ± 4.4	13.1 ± 3.9	0.842
Current BMI (kg/m^2^)	23.4 ± 3.5	23.2 ± 3.9	0.789
**Years of education**			
≤12	19 (20)	5 (22)	0.817
≥16	79 (80)	18 (78)	
**Working state**			
Homemaking	31 (32)	11 (48)	0.529
Full-time	49 (51)	9 (39)	
Part-time	16 (17)	3 (13)	
**Parity**			
Primipara	60 (62)	15 (65)	0.765
Multipara	37 (38)	8 (35)		
**Baby’s gender**			
Male	58 (60)	12 (52)	0.505	
**Type of delivery**			
Vaginal	67 (69)	13 (57)	0.251
Caesarian section	30 (31)	10 (43)	
**Baby’s caregiver**			
Self	72 (75)	15 (65)	0.342
Family members	24 (25)	8 (35)	
**Breast-feeding**			
Exclusive breast feed	34 (35)	6 (26)	0.533
Mix feed	53 (55)	13 (57)	
Formula feed	10 (10)	4 (17)	
**Nutritional supplements**			
Pregnancy first trimester	88 (91)	23 (100)	0.129
Pregnancy second trimester	82 (85)	19 (83)	0.820
Pregnancy third trimester	78 (81)	18 (78)	0.817
Lactation	40 (41)	9 (39)	0.853
**Postpartum confinement**			
Home care	44 (45)	16 (70) *	0.037
Postpartum care center	53 (55)	7 (30)	
**S** **elf-perceived psychological stress**			
The ability to care for the baby	49 (51)	15 (65)	0.204
Families unable to provide support	8 (8)	7 (30) *	0.004
Husband unable to provide support	1 (1)	5 (22) *	0.000
Self-health status	8 (8)	5 (22)	0.061
Baby health status	35 (36)	10 (44)	0.510
Financial ability	15 (16)	4 (17)	0.820
Other	3 (3)	3 (13) *	0.049
**Overall psychological stress scores**	1.2 ± 1.0	2.1 ± 1.0*	0.000
**Postpartum care satisfaction scores**	6.1 ± 1.0	5.2 ± 1.2*	0.005

*: Significant difference between postpartum depressive symptom (PPDS) and non-PPDS groups as analyzed by an independent Student’s t test or chi-squared test (*p* < 0.05).

**Table 2 nutrients-11-01204-t002:** Nutritional biomarkers of the postpartum women ^a^.

	Non-PPDS	PPDS	
	*N* = 97	*N* = 23	*p* Value
**HGB (g/dL)**	13.0 ± 1.2	13.3 ± 0.9	0.363
**Ferritin (ng/mL)**	58.0 ± 51.9	57.7 ± 48.4	0.977
**25(OH)D3 (ng/mL)**	24.0 ± 8.0	26.3 ± 7.1	0.197
**Retinol (μmol/L)**	1.3 ± 0.4	1.5 ± 0.5	0.050
**α-tocopherol (μmol/L)**	26.0 ± 6.6	24.9 ± 8.6	0.496
**Riboflavin (ng/mL)**	7.2 ± 2.0	6.2 ± 2.1 *	0.036

^a^ Values are presented as means ± SDs. *: Significant differences between postpartum depressive symptom (PPDS) and non-PPDS groups as analyzed by an independent Student’s t test (*p* < 0.05).

**Table 3 nutrients-11-01204-t003:** Fatty acids composition of erythrocytes ^a^.

Fatty Acids	Non-PPDS	PPDS	
% of total FAs	***N* = 97**	***N* = 23**	***p* value**
**Total saturated fatty acids**	41.45 ± 0.81	41.59 ± 0.63	0.432
C12:0	0.08 ± 0.02	0.08 ± 0.02	0.492
C14:0	0.17 ± 0.05	0.17 ± 0.04	0.986
C15:0	0.09 ± 0.02	0.09 ± 0.02	0.875
C16:0	22.63 ± 0.70	22.62 ± 0.50	0.926
C17:0	0.27 ± 0.03	0.27 ± 0.03	0.940
C18:0	14.41 ± 0.75	14.48 ± 0.54	0.740
C20:0	0.30 ± 0.04	0.30 ± 0.03	0.699
C22:0	1.09 ± 0.12	1.09 ± 0.12	0.939
C24:0	2.40 ± 0.29	2.50 ± 0.32	0.971
**Total monounsaturated fatty acids**	17.50 ± 0.79	17.37 ± 0.60	0.469
C14:1n-5	0.04 ± 0.03	0.04 ± 0.02	0.757
C16:1n-9	0.08 ± 0.01	0.07 ± 0.01*	0.007
C16:1n-7	0.25 ± 0.11	0.25 ± 0.04	0.745
C18:1n-9	12.46 ± 0.64	12.47 ± 0.50	0.933
C18:1n-7	1.02 ± 0.08	1.01 ± 0.11	0.668
C20:1n-9	0.27 ± 0.04	0.28 ±0.04	0.359
C24:1n-9	3.38 ± 0.28	3.24 ±0.22*	0.032
**Total polyunsaturated fatty acids**			
**Σn-6**	30.35 ± 1.45	30.97 ± 1.16	0.059
C18:2n-6	11.14 ± 1.47	11.28 ± 0.98	0.668
C18:3n-6	0.05 ± 0.02	0.04 ± 0.02*	0.001
C20:2n-6	0.48 ± 0.05	0.47 ± 0.05	0.684
C20:3n-6	1.24 ± 0.19	1.34 ± 0.22*	0.030
C20:4n-6	14.25 ± 1.01	14.44 ± 0.83	0.408
C22:4n-6	2.63 ± 0.38	2.77 ± 0.50	0.134
C22:5n-6	0.56 ± 0.12	0.63 ± 0.14*	0.013
**Σn-3**	10.07 ± 1.06	9.50 ± 0.93*	0.020
C18:3n-3	0.15 ± 0.03	0.15 ± 0.04	0.100
C20:5n-3 (EPA)	1.61 ± 0.43	1.34 ± 0.34*	0.006
C22:5n-3	1.94 ± 0.22	1.91 ± 0.21	0.543
C22:6n-3 (DHA)	6.40 ± 0.76	6.11 ± 0.72	0.102
**Σn-6/** **Σn-3 ratio**	3.05 ± 0.42	3.30 ± 0.44*	0.015
**C20:3n-6/C18:3n-6**	30.57 ± 23.42	54.93 ± 36.76*	0.005
**C20:4n-6/(C20:5n-3 + C22:6n-3)**	1.81 ± 0.27	1.97 ± 0.29*	0.012
**C22:6n-3/C22:5n-6**	11.90 ± 2.88	10.05 ± 2.33*	0.005

^a^ Values are presented as means ± SDs. * Mean significant difference between postpartum depressive symptom (PPDS) and non-PPDS groups as analyzed by independent Student’s t tests (*p* < 0.05).

**Table 4 nutrients-11-01204-t004:** Association between nutritional biomarkers and postpartum depression.

	β	S.E ^a^	OR	*p* Value	95% CI
**25(OH)D_3_ (ng/mL)**					
Model 1^b^	0.037	0.029	1.038	0.198	0.981–1.099
Model 2	0.031	0.033	1.032	0.352	0.966–1.101
**Retinol (μmol/L)**					
Model 1	0.998	0.517	2.714	0.054	0.985–7.480
Model 2	1.207	0.618	3.343	0.051	0.996–11.228
**α-tocopherol (μmol/L)**					
Model 1	−0.024	0.035	0.976	0.493	0.912–1.045
Model 2	−0.013	0.040	0.987	0.739	0.912–1.068
**Riboflavin (ng/mL)**					
Model 1	−0.291	0.142	0.747*	0.040	0.566–0.987
Model 2	−0.379	0.157	0.684*	0.015	0.504–0.930

^a^ S.E = standard error of the mean; OR = odds ratio; 95% CI = 95% confidence interval. *: Significant difference between postpartum depressive symptom (PPDS) and non-PPDS groups as analyzed by binary logistic regression (*p* < 0.05). ^b^ Model 1: not adjusted; Model 2: adjusted for the type of postpartum confinement, overall psychological stress scores, and postpartum care satisfaction scores.

**Table 5 nutrients-11-01204-t005:** Association between fatty acid composition of erythrocyte and postpartum depression ^a^.

	β	S.E	OR	*p* Value	95% CI
**Total saturated fatty acids**					
Model 1 ^b^	0.002	0.003	1.002	0.432	0.997–1.008
Model 2	0.003	0.003	1.003	0.364	0.997–1.009
**Total monounsaturated fatty acids**					
Model 1	−0.002	0.003	0.998	0.466	0.992–1.004
Model 2	−0.009	0.004	0.991*	0.003	0.983–0.999
**C16:1n-9**					
Model 1	−0.522	0.200	0.593*	0.009	0.401–0.878
Model 2	−0.686	0.241	0.504*	0.004	0.314–0.808
**C24:1n-9**					
Model 1	−0.019	0.009	0.981*	0.036	0.964–0.999
Model 2	−0.034	0.012	0.967*	0.006	0.943–0.990
**n-6 Polyunsaturated fatty acids**					
**Σn-6**					
Model 1	0.004	0.002	1.004	0.055	1.000–1.008
Model 2	0.005	0.002	1.005*	0.038	1.000–1.009
**C18:3n-6**					
Model 1	−0.386	0.125	0.680*	0.002	0.532–0.869
Model 2	−0.461	0.148	0.630*	0.002	0.472–0.842
**C20:3n-6**					
Model 1	0.025	0.012	1.025*	0.036	1.002–1.050
Model 2	0.021	0.015	1.022	0.141	0.993–1.051
**C22:5n-6**					
Model 1	0.042	0.018	1.043*	0.018	1.007–1.079
Model 2	0.043	0.021	1.044*	0.045	1.001–1.089
**n-3 Polyunsaturated fatty acids**					
**Σn-3**					
Model 1	−0.006	0.002	0.994*	0.023	0.990–0.999
Model 2	−0.004	0.003	0.996	0.155	0.991–1.001
**C20:5n-3**					
Model 1	−0.020	0.008	0.980*	0.008	0.966–0.995
Model 2	−0.019	0.008	0.982*	0.020	0.966–0.997
**C22:6n-3**					
Model 1	−0.005	0.003	0.995	0.104	0.989–1.001
Model 2	−0.003	0.004	0.997	0.356	0.990–1.004
**Σn-6/Σn-3**					
Model 1	0.013	0.005	1.013*	0.019	1.002–1.024
Model 2	0.010	0.006	1.010	0.113	0.998–1.023
**C20:3n-6/C18:3n-6**					
Model 1	0.025	0.008	1.025*	0.001	1.010–1.041
Model 2	0.031	0.009	1.031*	0.001	1.014–1.049
**C20:4n-6/(** **C20:5n-3 + C22:6n-3)**					
Model 1	0.022	0.009	1.022*	0.015	1.004–1.040
Model 2	0.020	0.010	1.020*	0.048	1.000–1.041
**C22:6n-3/C22:5n-6**					
Model 1	−0.270	0.100	0.764*	0.007	0.627–0.930
Model 2	−0.241	0.116	0.786*	0.037	0.626–0.986

^a^ 95% CI = 95% confidence interval; OR = odds ratio. *: Significant difference between postpartum depressive symptom (PPDS) and non-PPDS groups as analyzed by binary logistic regression (*p* < 0.05). ^b^ Model 1: not adjusted; Model 2: adjusted for the type of postpartum confinement, overall psychological stress scores, and postpartum care satisfaction scores.

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
