# Peer review of "Association between Postpartum Nutritional Status and Postpartum Depression Symptoms"

_nutrients, 2019, doi:10.3390/nu11061204_

Round 1

Reviewer 1 Report

Many thanks to the authors and the editor for allowing me to evaluate this interesting work.

This paper describes a study that investigate the association between nutritional status and postpartum depression at 6–8 weeks postpartum. The article is properly written, following all the guidelines for publications of scientific articles.

Introduction

The introduction is very short, I would like to know how is the status of the issue in the rest of the world, ask the authors to expand this information, and not only focus on China or Taiwan.

Methods

The method is perfectly described by presenting the sociodemographic variables, the place, and the way of performing the simulations, and how everything done through the corresponding statistical tests was evaluated later.

 The article has gone through an ethical committee having its approval.

Results  

The tables and data presented are correct.

Discussion

The discussion is perfectly organized.

Author Response

Response to Reviewer 1 Comments

Point: Many thanks to the authors and the editor for allowing me to evaluate this interesting work. This paper describes a study that investigate the association between nutritional status and postpartum depression at 6–8 weeks postpartum. The article is properly written, following all the guidelines for publications of scientific articles.

Introduction

The introduction is very short, I would like to know how is the status of the issue in the rest of the world, ask the authors to expand this information, and not only focus on China or Taiwan.

Methods

The method is perfectly described by presenting the sociodemographic variables, the place, and the way of performing the simulations, and how everything done through the corresponding statistical tests was evaluated later. The article has gone through an ethical committee having its approval.

Results  

The tables and data presented are correct.

Discussion

The discussion is perfectly organized.

Response:

We greatly appreciate reviewer’s thoughtful comments that helped improve the manuscript. We expand two useful references for PPD information in Introduction (line 54). Journal of Psychosomatic Research 2000, 49, 207-216. (line 358) and BMC public health 2011, 11, 302-302. (line 361).

Reviewer 2 Report

The occurrence of postpartum depression (PPD) is an increasingly frequent problem, with multifactorial background, which is widely documented in the literature. One of the factors is nutritional deficiencies. The aim of the present study was to investigate the association between nutritional status and postpartum depression at 6–8 weeks postpartum. The manuscript concerns an important and current topic. Moreover, it contains all the required sections and each of them is comprehensivly described, so the paper seems to be well done, the experiments were properly designed and whole data have outstanding scientific value.

Presented work contains 14 typewritten pages including 32 references thematically related to the content of the paper. The Abstract is compendious. There are only some minor editing errors which should be carefuly amended. The study have also some limitations, which are honestly indicated by the Authors.

I do not have any more comments.

Author Response

Response to Reviewer 2 Comments

Point: The occurrence of postpartum depression (PPD) is an increasingly frequent problem, with multifactorial background, which is widely documented in the literature. One of the factors is nutritional deficiencies. The aim of the present study was to investigate the association between nutritional status and postpartum depression at 6–8 weeks postpartum. The manuscript concerns an important and current topic. Moreover, it contains all the required sections and each of them is comprehensivly described, so the paper seems to be well done, the experiments were properly designed and whole data have outstanding scientific value.

Presented work contains 14 typewritten pages including 32 references thematically related to the content of the paper. The Abstract is compendious. There are only some minor editing errors which should be carefuly amended. The study have also some limitations, which are honestly indicated by the Authors.

I do not have any more comments.

Response: We greatly appreciate reviewer’s thoughtful comments that helped improve the manuscript. In order to consistent in manuscript, the abbrevation of PPDSs was replaced by PPDS in updated version (line 31, line 32, line 37, line 41, line 67, line 102, line 200, line 211, line 215, line 217, line 225, line 228, line 235, line 244, line 273, line 275, line 298, line 299, line 322, line 335, and line 346.

Reviewer 3 Report

The work of Lin et al is quite interesting and it cobers an topic regarding the environmenatl risk factors in post-partum depression. However, literature reported is not updated and no mechanism has been hypothesized in the study. In particular, they refer to serotoni reduction in condition of low n-3 PUFA consumption, In this regard many novel findings very recent can be added to their discussion in order support their hypothesis (Biochem Pharmacol. 2018 Sep;155:326-335; Mol Neurobiol. 2017 Apr;54(3):2079-2089; J Nutr Biochem. 2018 Aug;58:37-48. and many other....). Furthermore, I would change tab 5 into a figure to better appreciate the effects found.

Author Response

Response to Reviewer 3 Comments

Point: The work of Lin et al is quite interesting and it cobers an topic regarding the environmenatl risk factors in post-partum depression. However, literature reported is not updated and no mechanism has been hypothesized in the study. In particular, they refer to serotoni reduction in condition of low n-3 PUFA consumption, In this regard many novel findings very recent can be added to their discussion in order support their hypothesis and possible mechanism.  (Biochem Pharmacol. 2018 Sep;155:326-335; Mol Neurobiol. 2017 Apr;54(3):2079-2089; J Nutr Biochem. 2018 Aug;58:37-48. and many other....).

Response: We greatly appreciate reviewer’s thoughtful comments that helped improve the manuscript. The expand three useful references related to serotonin reduction in condition of low n-3 PUFA consumption was discussed to support possible mechanism in updated version (line 304-307) including Biochemical Pharmacology 2018, 155, 326-335. (line 449), Mol Neurobiol 2017, 54, 2079-2089. (line 452) and The Journal of Nutritional Biochemistry 2018, 58, 37-48. (line 455).

Point: Furthermore, I would change tab 5 into a figure to better appreciate the effects found.

Response: Due to logistic regression applied in postpartum depression (PPD) usually presented as table in manuscript such as (Prostaglandins, Leukotrienes and Essential Fatty Acids 69 (2003) 237–243; Health Care for Women International. (2011) 32:939–949; Front. Psychiatry 8:241. (2017) doi: 10.3389/fpsyt.2017.00241. etc.). Therefore, we would stay the same in table 5 would be more clear and helpful to readers unfamiliar with the subject.
